# Incomplete learning of multi-modal connectome for brain disorder diagnosis via modal-mixup and deep supervision

**Yanwu Yang**[1,2]                                          20B952019@stu.hit.edu.cn
**Hairui Chen**[3]                                              hrychen@163.com
**Zhikai Chang**[1]                                            22B352010@stu.hit.edu.cn
**Yang Xiang**[2]                                              xiangyang.hitsz@gmail.com
**Chenfei Ye**[*2,5]                                           chenfei.ye@foxmail.com
**Ting Ma**[*1,,4,5]                                           tma@hit.edu.cn

[1] *Electronic and Information Engineering School, Harbin Institute of Technology (Shenzhen), Shenzhen, China*

[2] *Peng Cheng Laboratory, Shenzhen, China* [3] *Beijing University Of Posts and Telecommunications, Beijing, China*

[4] *Guangdong Provincial Key Laboratory of Aerospace Communication and Networking Technology, Harbin Institute of Technology (Shenzhen), Shenzhen, China*

[5] *International Research Institute for Artificial Intelligence, Harbin Institute of Technology (Shenzhen), Shenzhen, China*

**Editors:** Accepted for publication at MIDL 2023

## Abstract

Recently, the study of multi-modal brain networks has dramatically facilitated the efficiency in brain disorder diagnosis by characterizing multiple types of connectivity of brain networks and their intrinsic complementary information. Despite the promising performance achieved by multi-modal technologies, most existing multi-modal approaches can only learn from samples with complete modalities, which wastes a considerable amount of mono-modal data. Otherwise, most existing data imputation approaches still rely on a large number of samples with complete modalities. In this study, we propose a modal-mixup data imputation method by randomly sampling incomplete samples and synthesizing them into complete data for auxiliary training. Moreover, to mitigate the noise in the complementary information between unpaired modalities in the synthesized data, we introduce a bilateral network with deep supervision for improving and regularizing mono-modal representations with disease-specific information. Experiments on the ADNI dataset demonstrate the superiority of our proposed method for disease classification in terms of different rates of samples with complete modalities.

**Keywords:** Brain connectome, incomplete learning, deep supervision, brain disorder, missing modalities

## 1. Introduction

Neuroimaging developments have been widely adopted in medical scenarios such as disease diagnosis (Zhu et al., 2021; Liu et al., 2022; Yang et al., 2022) and lesion segmentation (Falk et al., 2019; Menze et al., 2014; Zhou et al., 2018). Advanced neuroimaging candidates, i.e., functional Magnetic Resonance Imaging (fMRI) and Diffusion Tensor Imaging (DTI),

---

[*] Corresponding authors

are powerful tools for brain disorder diagnosis by characterizing neural connections and information flow between brain regions (Yin et al., 2022; Li et al., 2021; Kawahara et al., 2017). Derived functional and structural connectomes are modeled as graphs by representing the activity of neurons as nodes interconnected by a set of edges, providing a more holistic view for relating abnormal discharge of neurons and brain dysfunction (Dadi et al., 2019). Analysis of brain connectome can contribute to the scientific understanding of the cognitive process and potentially aid in diagnosing and treating neurological disorders (Gabrieli et al., 2015; Pu et al., 2015).

In general, the brain networks can be categorized into two classes: functional networks derived from fMRI or EEG, and structural networks obtained from DTI or DSI (Bullmore and Bassett, 2011; Achard et al., 2006; Zalesky and Fornito, 2009). Combining functional and structural connectome enables the exploration of brain state by neuron activation and connection in vivo by leveraging complementary information between functional and structural networks. And multi-modal brain networks provide a more constructive scene with distinctive biomarkers. Despite the promising performance achieved by multi-modal technologies, one of the core issues is that most methods can only use data with complete modalities. In practice, it is difficult to gather a large amount of complete data. Especially when diagnosing neurodegenerative diseases based on fMRI and DTI, only partial subjects have both images. In addition, missing modality is a common issue in real-world multi-modal scenarios, and the missingness can be caused by various reasons such as sensor damage, data corruption, and human mistakes in recording (Ding et al., 2018; Chen and Zhang, 2020). Unfortunately, most existing multi-modal methods may have to discard a large portion of the data collected.

One way to address the abovementioned issue is by incomplete learning (Ma et al., 2021; Liu et al., 2021; Ding et al., 2021). Imputation-based approaches fill incomplete missing modalities (Wang et al., 2020) with training data by advanced generative models such as generative adversarial networks (GAN) (Goodfellow et al., 2020) and leverage multi-modal learning methods to embed multi-modal representations. Knowledge distillation-based approaches guide the student model training with regularized mono-modal latent knowledge from a pretrained teacher model. However, these approaches still depend on the amount of samples with complete modalities. Learning with insufficient samples may lead to model over-fitting and poor generalization, especially when there exist severe missing modalities.

In this regard, we propose a novel data imputation approach, modal-mixup, for synthesizing samples with incompleteness into complete data for training. The idea behind this is by mix-up data augmentation approach (Zhang et al., 2017; Verma et al., 2019; Thulasidasan et al., 2019), where samples of different classes are randomly mixed into updated samples. The modal-mixup method randomly samples data with missing modalities to constitute complete data. On another hand, the imputation methods may introduce extra noise to the inter-modal complementary information. Learning with unpaired structural and functional connectome has negative impacts on performance, especially for most existing multi-modal networks that apply complementary information for learning. Accordingly, we propose a bilateral deep-supervision representation learning network to avoid learning with inter-modal dependency. The deep-supervision is introduced as a regularization term to reinforce and improve the latent mono-modal features with disease-specific

semantics for classification. We evaluated the proposed modal-mixup and bilateral network on the ADNI database with a cohort of 124 subjects. Our code is available online at https://github.com/podismine/IncompleteModality.

The contributions are summarized as follows:

- A modal-mixup data imputation approach is proposed for synthesizing samples with incompleteness into complete data to mitigate the need for numerous complete data.

- We introduce a bilateral deep-supervision representation learning network to regularize the latent representations with disease-specific semantics instead of learning with inter-modal dependencies.

- Extensive experiments have been performed to validate the superiority of our proposed approach in improving the classification performance of incomplete learning.

## 2. Related works

**Multi-modal Brain Connectome study.** Recently, deep learning methods have been state-of-the-art tools to embed high-order representations and achieve promising performances. For example, (Wang et al., 2018) performed a multi-layer convolution on fMRI and DTI data simultaneously. (Dsouza et al., 2021) regularized convolution on functional connectivity with structural graph Laplacian. A triplet attention network with a self-attention mechanism was introduced to map high-order multi-modal representations (Zhu et al., 2022). (Feng et al., 2019) proposed to perform hyperedge to perform heterogeneous graph convolution on multi-modal data. However, these approaches have rarely explored learning with complementary information from incomplete multi-modal data.

**Incomplete Learning.** The exploration of incomplete learning with missing modalities has recently attracted much attention. In most cases, incomplete learning can be divided into two conditions: the test set is complete and incomplete, while the training samples are incomplete with missing modalities. In this study, we focus on studying with the complete test set. In this regard, more mono-modal neuroimages, which are easier to be collected, could be gathered for training, which might improve the model performance and generalizability. Besides, data imputation methods such as KNN are commonly used in most conditions (Campos et al., 2015). Advanced imputation methods, such as adversarial training with a similar structure as GAN, have also been proposed to deal with imputing the missing modalities (Cai et al., 2018). (Wang et al., 2020) proposed a knowledge distillation-based approach to integrating the supplementary information of multiple modalities. However, most of these studies still rely on a large amount of complete data to obtain a well-trained data imputation model.

## 3. Method

### 3.1. Problem formulation

The training sets included complete data and incomplete data, while the test sets were complete. For a multi-modal dataset with $M$ modalities, there are $2^M - 1$ different combinations of missing modalities. In this study, functional and structural brain networks are

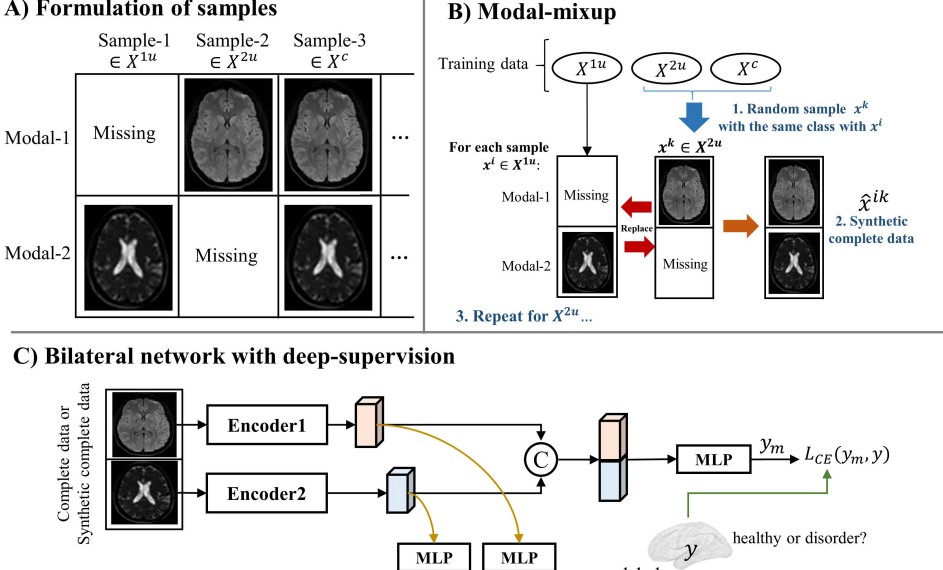

Figure 1: Illustration of A) the data samples, B) proposed modal-mixup, and C) bilateral network with deep supervision.

studied with $M = 2$. Especially for the training sets, the samples with missing modalities are denoted as $X^{1u} \in R^{n_{1u} \times d_{1u}}, X^{2u} \in R^{n_{2u} \times d_{2u}}$. And the complete data are represented as $X^c \in R^{n_c \times d_c}$ with a total of $N_{train} = n_c + n_{1u} + n_{2u}$ training samples, as shown in Figure 1 A). In this study, given a collection of incomplete multi-modal data samples $\{X_i\}_{i=1}^N$ as input, where each sample consists of a set of available modalities $X_i = \{x_{i,m}\}$, our goal is to design a model to capture dependencies between modalities and fuse multi-modal data with missing modalities in an architecture.

### 3.2. Modal-Mixup

([Zhang et al., 2017](#)) first proposed the Mixup method for image classification, where synthetic samples are generated by linearly interpolating a pair of training samples as well as their targets. Consider a pair of samples $(x^i; y^i)$ and $(x^j; y^j)$, a synthetic sample is generated as $\hat{x}^{ij} = \lambda x^i + (1-\lambda)x^j, y^{ij} = \lambda y^i + (1-\lambda)y^j$, where $\lambda \in (0,1)$ is the mixing ratio for the pair and $y^i$ and $y^j$ are one-hot label encodings. In this study, we follow the mixup approach and propose a modal-mixup method by replacing the combined operations with concatenation layers. Moreover, the concatenation of modalities assumes non-linear combinations of the associated targets. In this regard, we propose to sample the random data with the same class. In detail, the synthetic data samples are obtained by:

$$
\begin{aligned}
&\text{For } x^i \in X^{1u}, \hat{x}^{ik} = ||\{x^i, x^k\}, x^k \in X^{2u} \cup X^c, y^i = y^k \\
&\text{For } x^j \in X^{2u}, \hat{x}^{jk} = ||\{x^k, x^j\}, x^k \in X^{1u} \cup X^c, y^j = y^k
\end{aligned}
\tag{1}
$$

where $||$ denotes the concatenation operation. The updated label is obtained as the same class. As is shown in Figure 1 B), for each incomplete data $x^i \in X^{1u}$, we randomly sample a case $x^k$ from the data with modality 1 missing $x^k \in X^{2u} \cup X^c$ with the same class with $x^i$, $y^i = y^k$. The synthesized data are updated by concatenation as Eq. (1). In this regard, the samples with modal-1 missing are mixed with those with modal-2 missing and constitute complete data $\hat{x}^{ik}$. This step is repeated for all incomplete samples in $X^{1u}$ and $X^{2u}$.

### 3.3. Bilateral network with deep supervision

As is shown in Figure 1 C), our network is embedded with bilateral encoders for parsing heterogeneous multi-modal brain network representations. In detail, a two-layer multi-layer perception (MLP) is used for each branch. And the MLP layers take the vectorized brain connectome features into 32 features, followed by ReLU activation and dropout. The parsed representations are further concatenated and fed into a classifier.

In addition, one core challenge of data imputation by modal-mixup mentioned above is the unpaired inter-modal complementary information that would introduce unwanted noise and decrease performance. In order to reduce the effect of interactions of inter-modal representations, in this study, we propose to discard inter-modal dependency learning modules proposed by other studies. Moreover, we improve the mono-modal representations and regularize them with disease-specific information for classification. In detail, the intermediate representations of the bilateral branches are fed into a 3-layer MLP for prediction. The deep-supervision loss $L_{DS}$ is represented by a summation of two cross-entropy losses of the deep-supervision outputs $y_s = \{y_s | s = 1, 2\}$ and the target $y$ as $L_{DS} = \sum_{s=1}^{m} L_{CE}(y_s, y)$.

### 3.4. Optimization

In the training process, the synthetic data and complete samples are fed into the framework. The objective function is constructed by a weighted combination of cross-entropy loss from the target output and the deep supervision as:

$$L_{DS} = L_{target} + L_{DS} = L_{CE}(y_m, y) + \lambda \sum_{s=1}^{m} L_{CE}(y_s, y) \tag{2}$$

where $L_{CE}(y_m, y)$ denotes the cross-entropy loss between multi-modal prediction outputs $y_m$, and the ground truth and $L_{CE}(y_s, y)$ represents that between the $s$-th deep supervision output and the ground truth.

## 4. Experiments

### 4.1. Datasets and preprocessing

In this study, the ADNI database (http://www.adni-info.org/) was used to form the cohort, where fMRI and DTI images are collected. In this study, we collected 124 subjects that were diagnosed at the baseline for evaluation, including 61 healthy controls (NC) and 63 with mild cognitive impairment (MCI). Notably, MCI is considered to be a significant

stage for the preclinical diagnosis of AD. The patients were diagnosed at baseline, and the HCs were healthy at their first examination.

All the fMRI images were pre-processed by reference to the Configurable Pipeline for the Analysis of Connectomes (CPAC) pipeline (Craddock et al., 2013), including skull striping, slice timing correction, motion correction, global mean intensity normalization, nuisance signal regression with 24 motion parameters, and band-pass filtering (0.01-0.08Hz). The functional images were finally registered into standard anatomical space (MNI152). The mean time series for a set of regions were computed and normalized into zero mean and unit variance. Pearson Coefficient Correlation was applied to measure functional connectivity. The DTI images were pre-processed by image denoising, head motion, eddy-current, susceptibility distortion, and field inhomogeneity correction by MRtrix 3 (Tournier et al., 2012). The streamline count was reconstructed to 5 million. In this study, the fMRI and DTI images were segmented by the Schaefer atlas (Schaefer et al., 2018) that identified 100 cortical parcels.

### 4.2. Implementation details

In our implementation, the number of layers for deep supervision is 3, with an output size of 128, 30, and 2, respectively. Each layer is followed by a leaky ReLU activation function and a dropout layer. The learning rate is set as 3e-4, and the weight decay is 5e-5. The weight $\lambda$ in the loss function is set as 1. All the models in this study are trained for 600 epochs and would be stopped early when the loss has not been decreased for 50 epochs. We trained the models with PyTorch on one NVIDIA 2080-Ti GPU. 10-fold cross-validation was applied for evaluation, where 10% samples were randomly selected for testing for each fold. For all experiments, we evaluated the performance in terms of diagnosis accuracy (Acc), sensitivity (Sen), and F1-score (F1).

### 4.3. Competitive baseline

In this study, we compare our proposed method with different combinations of data imputation approaches with various multi-modal learning frameworks. The data imputation methods include training with only complete data (C), missing modality imputation by K-nearest neighbors (KNN) (Campos et al., 2015), adversarial-based imputation (ADV) (Cai et al., 2018), knowledge distillation-based information integration (KD) (Wang et al., 2020) and missing modality imputation with averaged complete data (Mean). And the competitive multi-modal networks involve BrainNetCNN (Kawahara et al., 2017), Triplet Attention Network (TAN) (Zhu et al., 2022), M-GCN (Dsouza et al., 2021), HGCN (Feng et al., 2019).

## 5. Results

### 5.1. Comparison results on incomplete learning

Figure 2 demonstrates the results of competitive networks with different data imputation approaches in various ratios of complete data. The results of TAN with KD-based imputation are missing since the network is proposed for multi-modal data, which is not available for the KD-based method. In the figure, the results of MLP, BrainNetCNN, MGCN, HGCN,

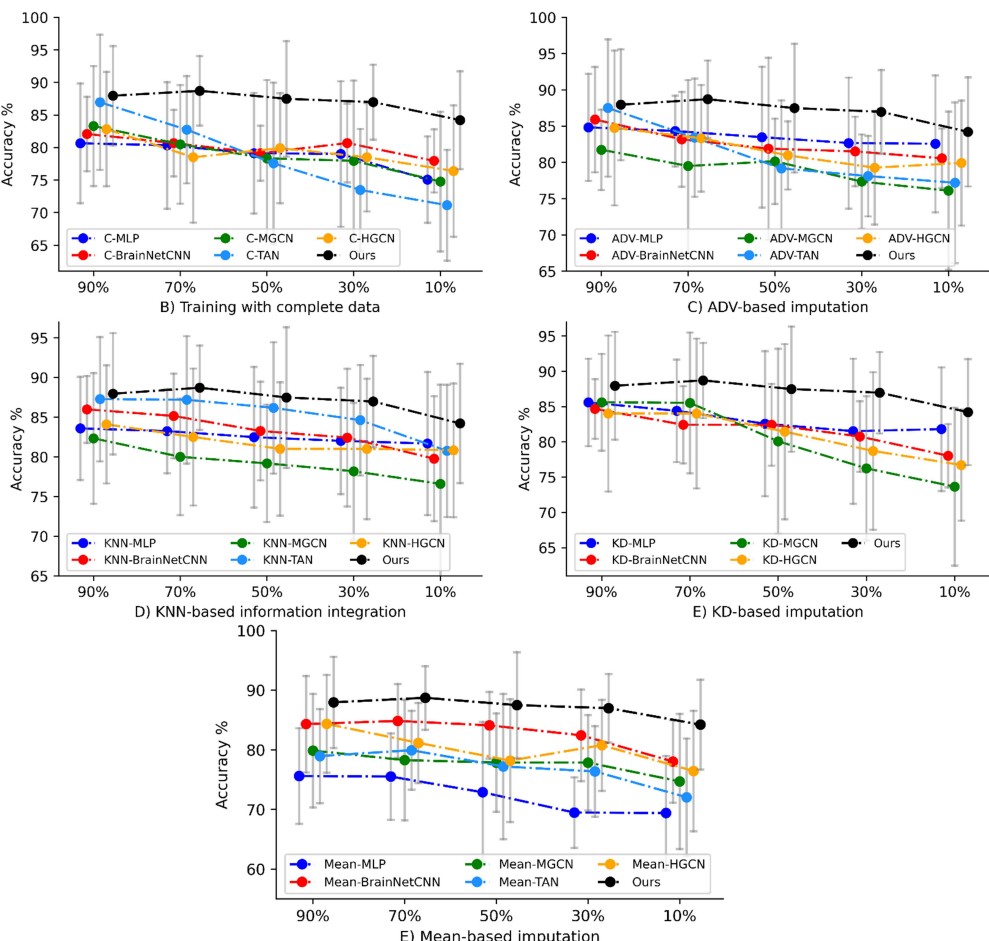

Figure 2: Classification accuracy of five models by training with A) complete data, B) adversarial-based imputation, C) KNN-based imputation, D) KD-based imputation, and E) Mean-based imputation. Our proposed method is shown in a solid black line in all four figures for comparison. The variations are shown in grey.

and TAN are shown in dark blue, red, green, light blue, and orange, respectively, while those of our bilateral framework are shown in black. It can be seen that as the rate of complete data decreases, the performance becomes smaller. To note that, as fewer complete samples are used for training, the performance of these methods does not always degrade. For example, training with 70% complete data by BrainNetCNN performs worse than that with 50% complete data. We hypothesize that the number of samples enrolled in our study is relatively small, and there might be fluctuations.

In addition, from Figure 2 A), we can see that TAN performs the best with more than 50% complete data among five well-estimated networks while achieving the worst results when there is severe incomplete data, i.e., the ratio of complete data < 50%. In this regard, TAN depends more on the number of training samples. Figure 2 B), C),

D), and E) display the results of the networks using adversarial-based, KNN-based, KD-based approaches, and mean-based approaches. In the mean-based imputation method, the five networks generally outperform those trained with only complete data. Notably, the data imputation methods might bring in noise and decrease performance. From the results, we can see that in most cases there is a 5%± drop in accuracy with 10% complete data compared with 90% complete data, which is lower than that using other imputation methods such as ADV. Despite the relatively stable performance, the accuracy is still lower than our proposed approach. In particular, when there is only 10% data for training, MLP outperforms other architectures in most cases. This indicates that such highly complex multi-modal architectures depend on learning with complementary information between modalities and are limited to decoupling unpaired inter-modal associations. Moreover, compared with these approaches, our proposed method is feasible to achieve a slight drop in performance, even if there are severe incomplete samples.

Table 1 further demonstrates the detailed results with 10% complete data for training, where the accuracy (Acc), sensitivity (Sen), and F1-score (F1) are displayed. In most cases, networks with the KNN-based data imputation method achieve better performances than other approaches. This is because the other data imputation methods are still limited by the number of complete data. For example, the adversarial-based method depends on the complete data to generate accurate and robust data for missing modalities. Overall, even with only 10% complete data, our proposed method could achieve a promising performance with an accuracy of 84.21%, a sensitivity of 89.79%, and an F1-score of 84.21%.

Table 1: Comparison results on incomplete learning with 10% complete data in terms of accuracy (Acc %), sensitivity (Sen %), and F1-score (F1 %). The average and standard deviation (Mean±Std) across ten folds are displayed.

| Type | Model | ADNI Dataset | | |
|---|---|---|---|---|
| | | ACC | Sen | F1 |
| C | MLP | 75.08±6.64 | 75.66±7.94 | 75.08±6.64 |
| | BrainNetCNN | 77.95±4.84 | 81.67±12.62 | 77.95±4.84 |
| | M-GCN | 74.77±10.73 | 77.644±9.89 | 75.91±12.57 |
| | Triplenet | 71.14±8.51 | 76.17±9.92 | 71.14±8.51 |
| | HGCN | 76.44±10.09 | 74.38±9.47 | 80.35±9.18 |
| ADV | MLP | 82.58±9.44 | 81.13±8.82 | 85.15±8.11 |
| | BrainNetCNN | 80.56±4.11 | 79.80±7.03 | 80.56±4.11 |
| | M-GCN | 76.09±10.91 | 76.56±13.33 | 76.09±10.91 |
| | Triplenet | 77.20±11.08 | 77.35±14.84 | 81.90±7.21 |
| | HGCN | 79.92±8.61 | 79.18±12.67 | 79.92±8.61 |
| KNN | MLP | 81.67±9.02 | 81.93±10.25 | 83.99±8.48 |
| | BrainNetCNN | 79.77±7.87 | 88.10±10.43 | 79.77±7.87 |
| | M-GCN | 76.59±12.51 | 81.82±14.47 | 78.71±10.62 |
| | Triplenet | 80.76±8.33 | 82.23±10.63 | 80.76±8.34 |
| | HGCN | 80.83±8.43 | 84.52±11.39 | 80.83±8.43 |
| KD | MLP | 81.80±8.76 | 80.47±9.97 | 81.80±8.76 |
| | BrainNetCNN | 78.03±4.49 | 86.85±11.15 | 78.03±4.49 |
| | M-GCN | 73.64±11.20 | 81.43±15.97 | 73.64±11.20 |
| | HGCN | 76.71±7.88 | 84.83±9.50 | 76.71±7.88 |
| Mean | MLP | 69.39±9.58 | 74.05±15.09 | 69.39±9.58 |
| | BrainNetCNN | 78.03±6.93 | 82.61±13.30 | 78.03±6.92 |
| | M-GCN | 74.69±11.33 | 74.03±11.14 | 74.69±11.33 |
| | Triplenet | 72.04±9.83 | 83.27±14.77 | 72.05±9.83 |
| | HGCN | 76.44±10.09 | 74.38±9.47 | 76.44±10.09 |
| Modal-mixup | Deep-supervision (ours) | 84.21±7.52 | 89.79±10.84 | 84.21±7.52 |

## 5.2. Ablation studies

Table 2 demonstrates the results of ablation studies by training complete data with different ratios of complete data. From the results, we can see that although our method with modal-mixup outperforms other models, the deep supervision could further improve the accuracy by 1.56%, 3.56%, 3.09%, 3.56% and 1.81% with 10%, 30%, 50%, 70%, and 90% complete data. The results demonstrate the deep supervision facilitates to robust representations that are less dependent on the complementary information between modalities.

Table 2: Ablation studies on the modal-mixup and deep-supervision of different ratios of complete data. The average and standard deviation (Mean±Std) of accuracy across 10 folds are listed. The components evolved are shown in the table with ✓.

| Modal-mixup | deep-supervision | 10% | 30% | 50% | 70% | 90% |
|:---:|:---:|:---:|:---:|:---:|:---:|:---:|
| ✓ | | 82.65±9.98 | 83.41±5.76 | 84.39±9.85 | 85.15±5.34 | 86.14±7.64 |
| ✓ | ✓ | 84.21±7.52 | 86.97±6.79 | 87.48±8.87 | 88.71±6.55 | 87.95±10.01 |

## 6. Conclusion

In this study, we propose two strategies for incomplete learning with missing modalities: Modal-mixup is introduced for data imputation, which is less dependent on the amount of data with complete modalities; A bilateral network with deep-supervision is investigated for regularizing mono-modal representations. Experimental results on the ADNI dataset demonstrate that our proposed method is feasible to model incomplete data and outperforms other combinations of architectures and data imputation methods. Especially compared with 90% complete data, our proposed method could only achieve a 3.74% drop in terms of accuracy with 10% complete data. In this regard, our proposed method provides novel insights for multi-modal learning, i.e., in some medical scenarios more mono-modal samples can be gathered for multi-modal study to improve performance and generalizability.

## Acknowledgments

This study is supported by grants from the National Natural Science Foundation of P.R. China (62276081, 62106113), Innovation Team and Talents Cultivation Program of National Administration of Traditional Chinese Medicine (NO:ZYYCXTD-C-202004), Basic Research Foundation of Shenzhen Science and Technology Stable Support Program (GXWD20201230155427003-20200822115709001).

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

## Appendix A. Baseline Reproducibility

In this section, we first introduce the details of imputation approaches implemented in our experiments.

**Complete data**. The training data with complete modalities were used for training, and the incomplete samples were deleted.

**K-Nearest Neighbors**. For a sample with missing modalities, $k$ nearest neighbor samples with complete data in the training set were first sampled based on the distance of

the available modality. We filled the missing modality with the averaged features of the $k$ samples. The setting of $k$ is searched in the range of $(1, 5)$.

**Adversarial-based imputation**. We first trained generative models for reconstructing from modal 1 to modal 2 and modal 2 to modal 1 respectively based on the complete training data. The neural network architecture was implemented by using the backbone of (Cai et al., 2018). We replaced the original encoder for MRI with a combination of E2E, E2N, and N2G layers in BrainNetCNN. The missing modalities are further generated by pre-trained generative models.

**Knowledge-distillation**. For each modality, a teacher model is trained to learn mono-modal representations. The learned teacher knowledge is used to guide the student model training with pre-trained mono-modal knowledge. The teacher models take mono-modal data as inputs, while the student takes multi-modal as inputs. The KL divergence was implemented for knowledge distillation. More details can be found in (Wang et al., 2020). The DNN models are replaced by various models in our experiments.

The five neural network models (i.e. MLP, BrainNetCNN, MGCN, TAN, and HGCN) used in our study have public source code. We followed the originally proposed backbone and set the ROI number to 100. BrainNetCNN was implemented by multiple convolution layers to perform multi-modal data, where functional and structural connectivity matrices were concatenated by channel. For all the models, the settings of hyper-parameters such as hidden size and layer number are decided in a grid search.

