# OpenReview forum: "Incomplete learning of multi-modal connectome for brain disorder diagnosis via modal-mixup and deep supervision"
_MIDL.io/2023/Conference — MIDL 2023 Poster_

### Official Review · Reviewer_NCcQ · 2023-02-03

**Confidence:** 4
**Preliminary Rating:** 4

**Summary:**

The authors propose a method for multi-modal learning with incomplete modalities in MCI classification. They introduced a modality imputation strategy “model-mixup” which is inspired by mixup data augmentation technique. They also leverage bilateral networks and deep supervision for multimodal learning. In the experiments, fMRI and DTI images from ADNI are used for MCI v.s. HC classification. The proposed methods outperforms a list of baselines in incomplete learning setups.

**Strengths:**

- The paper is well motivated that modality incompleteness is a big problem for multi-view learning;
- The proposed method is straightforward;
- The authors compared the proposed method with many state-of-the-art baseline methods in the experiments;
- The experiments are thorough in terms of different incomplete learning conditions;


**Weaknesses:**

- In section 3.3, please elaborate on the deep-supervision loss and what s and m represent in the loss function;
- In section 3.4, why don't you need a balancing parameter between L_CE and L_DS?
- In section 4.1, why AD subjects from ADNI are not included in the experiments?
- In Figure 2, please show the standard variations from the 10-fold cross validation;


**Deanonymize Review:**

no

**Paper Type:**

methodological development

**Questions To Address In The Rebuttal:**

- Explain more on the deep supervision loss to help reader to understand;
- Explain why don’t need to balance the two loss functions;
- Explain why AD subjects were dropped;
- Add uncertainty measure to Figure 2;

---

### Official Review · Reviewer_XyLE · 2023-02-04

**Confidence:** 4
**Preliminary Rating:** 3

**Summary:**

The paper studies the problem of prediction with missing modality data for multimodal predictive connectomics. Specifically, they propose a strategy called modal mixup that imputes incomplete data by random sampling across the dataset to synthesize complete data. To mitigate the effects of noise thus arising, they introduce a bilateral network (based on concatenated features) with added supervision as a regularization. Experiments are performed on the ADNI dataset for disease classification under different rates of missing samples. Comparative analysis is provided against several deep learning models designed for prediction from multimodal/single modality connectomics (DTI+rs-fMRI) data.

**Strengths:**

The paper examines an important and prevalent problem in neuroimaging studies and in multimodal learning in general, that of prediction in the presence of incomplete (and high dimensional) data samples. Extensive comparisons are performed against several predictive deep learning models at various levels of missingness, using nearest neighbor based, adversarial and knowledge distillation based imputation strategies, against which the proposed approach appears to compare favorably.

**Weaknesses:**

Several points in the methodology need clarification:

1. A point that is not clear to me is how the predictions $y_{m}$ and $y_{s}$ are combined during inference to predict the final classification label, specifically during testing ( for computing evaluation metrics such as AUC, accuracy etc)

2. The explanation below Eq.(1) it reads as if $y_{ij}$ can be formed by mixing examples across different class labels. If this is indeed the case, I am not sure this is in general a meaningful label in general (take a three class classification with 0,1,2 as classes, mixing 0 and 2 this way gives rise to a label 1- which gives incorrect information during training). I am also not sure how the bilateral regularizer would address this.

**Deanonymize Review:**

no

**Detailed Comments:**

Minor comments:

Typos and grammatical issues:

1. Page 7 : hyperthesis -> hypothesize
2. Page 6: moodal -> modal

**Paper Type:**

methodological development

**Questions To Address In The Rebuttal:**

1. In addition to the points mentioned in weaknesses, it would be good to add a section explaining the baseline methods in detail (perhaps in the appendix) along with the implementation details.

2. Finally, have the authors tried to perform mean based imputation of unknown samples for any of the baselines- do their performance curves degrade as much as those with the other imputation strategies in Figure 2 for higher levels of missing data?

---

### Official Review · Reviewer_AHRi · 2023-02-06

**Confidence:** 4
**Preliminary Rating:** 3

**Summary:**

In this paper, the authors proposed a modal-mixup data imputation method by randomly sampling incomplete samples and synthesizing them into complete data for auxiliary training. Besides, a bilateral network with deep supervision is introduced for improving and regularizing mono-modal representations with disease-specific information, so as to mitigate the noise in the complementary information between unpaired modalities in the synthesized data. In general, it is well-written, and the experimental results are encouraging, but the novelty is limited and not clear.

**Strengths:**

1. A modal-mixup data imputation method is proposed by randomly sampling incomplete samples and synthesizing them into complete data for auxiliary training.
2. A bilateral network with deep supervision is introduced for improving and regularizing mono-modal representations with disease-specific information, so as to mitigate the noise in the complementary information between unpaired modalities in the synthesized data.
3. Experiments on the ADNI dataset demonstrate the superiority of the proposed method.

**Weaknesses:**

In general, it is well-written, and the experimental results are encouraging, but the novelty is limited and not clear. I have the following concerns:

1. The novelty of this paper is not introduced clearly. The authors should reorganize the manuscript and add more description to the Methods section to highlight the original contribution of the paper.

2. Figure 1 (b) does not clearly represent the proposed modal-mixup, and the proposed modal-mixup is a direct concatenation of two modal missing data, if it is data with different categories, then how did the authors solve the problem of modal category alignment?

3. Are authors planning to release the codes for the paper?

**Deanonymize Review:**

no

**Paper Type:**

both

**Questions To Address In The Rebuttal:**

1. The authors should reorganize the manuscript and add more description to the Methods section to highlight the original contribution of the paper.
2. Release the code to facilitate scholars in related research areas to follow the work

---

### Meta-Review · Area_Chair_D6od · 2023-02-24

**Recommendation:** Accept (Poster)
**Confidence:** 5

**Metareview:**

The authors have clearly clarified the issues raised by the reviewers. The explanations in the rebuttal look reasonable and correct to me.